# Development of CAPS Markers for Evaluation of Genetic Diversity and Population Structure in the Germplasm of Button Mushroom (*Agaricus bisporus*)

**DOI:** 10.3390/jof7050375

**Published:** 2021-05-11

**Authors:** Hyejin An, Hwa-Yong Lee, Donghwan Shim, Seong Ho Choi, Hyunwoo Cho, Tae Kyung Hyun, Ick-Hyun Jo, Jong-Wook Chung

**Affiliations:** 1Department of Industrial Plant Science and Technology, Chungbuk National University, Cheongju 28644, Korea; heayni@naver.com (H.A.); hwcho@chungbuk.ac.kr (H.C.); taekyung7708@chungbuk.ac.kr (T.K.H.); 2Department of Forest Science, Chungbuk National University, Cheongju 28644, Korea; leehy@chungbuk.ac.kr; 3Department of Biological Science, Chungnam National University, Daejeon 34134, Korea; dshim104@cnu.ac.kr; 4Department of Animal Science, Chungbuk National University, Cheongju 28644, Korea; seongho@cbnu.ac.kr; 5National Institute of Horticultural and Herbal Science, RDA, Eumseong 27709, Korea

**Keywords:** CAPS marker, genetic diversity, population structure, PCoA, AMOVA, accumulation curve, *Agaricus bisporus*

## Abstract

*Agaricus bisporus* is a globally cultivated mushroom with high economic value. Despite its widespread cultivation, commercial button mushroom strains have little genetic diversity and discrimination of strains for identification and breeding purposes is challenging. Molecular markers suitable for diversity analyses of germplasms with similar genotypes and discrimination between accessions are needed to support the development of new varieties. To develop cleaved amplified polymorphic sequences (CAPs) markers, single nucleotide polymorphism (SNP) mining was performed based on the *A. bisporus* genome and resequencing data. A total of 70 sets of CAPs markers were developed and applied to 41 *A. bisporus accessions* for diversity, multivariate, and population structure analyses. Of the 70 SNPs, 62.85% (44/70) were transitions (G/A or C/T) and 37.15% (26/70) were transversions (A/C, A/T, C/G, or G/T). The number of alleles per locus was 1 or 2 (average = 1.9), and expected heterozygosity and gene diversity were 0.0–0.499 (mean = 0.265) and 0.0–0.9367 (mean = 0.3599), respectively. Multivariate and cluster analyses of accessions produced similar groups, with *F*-statistic values of 0.134 and 0.153 for distance-based and model-based groups, respectively. A minimum set of 10 markers optimized for accession identification were selected based on high index of genetic diversity (GD, range 0.299–0.499) and major allele frequency (MAF, range 0.524–0.817). The CAPS markers can be used to evaluate genetic diversity and population structure and will facilitate the management of emerging genetic resources.

## 1. Introduction

Button mushroom (*Agaricus bisporus*) is a popular edible mushroom that is consumed worldwide. *A. bisporus* extracts have high antioxidant activity and are known to improve cardiovascular health [1,2]. Global mushroom production increased more than 30-fold during 1978–2013, and total production value increased to $63 billion [3]. *A. bisporus* production is approximately 4.4 million tons per year, constituting 15% of global mushroom production [3]. Despite the high economic value of button mushrooms, genetic diversity is low. Although new strains can be produced through phenotypic selection and limited parent strain crossing, high similarity remains among varieties [4]. Evaluation of the genetic characteristics of mushrooms with similar phenotypes is needed to facilitate the introduction of novel traits into commercial varieties. Molecular resources are also needed to support the efficient selection of accessions, collection and preservation of strains, diversity assessment, and population structure analysis [5]. The rapid development of next-generation sequencing technologies has enabled large-scale sequencing projects in a variety of organisms. Molecular markers that are stable, highly polymorphic, and provide valuable diversity information are preferred over phenotype-based markers that are subject to environmental effects [6]. Polymerase chain reaction (PCR)-based molecular markers, such as simple sequence repeats (SSR) and single nucleotide polymorphisms (SNPs), can be used for molecular genetic studies of culture collections. Genetic diversity in *A. bisporus* has been examined using several similar approaches, including analysis with restriction fragment length polymorphisms (RFLP) [7], discrimination analysis with random amplified polymorphic DNA (RAPD) [8], genetic diversity analysis using SSR markers [9,10,11], phylogenetic analysis using SNP markers [12], SNP genotyping [13,14], and quantitative trait locus (QTL) mapping using SNP markers [15,16]. However, molecular markers that can be used for the analysis of population structure and diversity of accession collections are biased toward SSR markers, and new SNP genotyping markers are needed.

SNPs, which are differences at single nucleotide positions between or within species, are the most widely distributed genetic variants in the genome. SNPs found in coding and non-coding regions are classified as transitions (Ts) (C/T or G/A) or transversions (Tv) (C/G, A/T, C/A, or T/G), depending on the type of nucleotide substitution. If present in coding regions, SNPs can change the structure or function of proteins, causing phenotypic differences [17]. Thus, compared with methods using other molecular markers, SNP genotyping-based diversity assessments can more accurately and specifically explain phenotypic differences. SNP markers are highly reproducible co-dominant markers that can be used to distinguish between homozygosity and heterozygosity and to discriminate accessions, and they can also be used for association mapping and analysis of genetic diversity and group structure [18,19,20]. Two main types of SNP-based markers are cleaved amplified polymorphic sequences (CAPS) and derived cleaved amplified polymorphic sequences (dCAPS), the latter of which offers increased utility for genotyping [21,22]. CAPS marker, also known as the PCR–RFLP marker, combined with primers that can amplify specific regions is popularly used for molecular genetic studies in fungus, owing to its advantage in detecting secondary polymorphisms that cannot be directly detected by PCR amplification [23,24]. Moreover, since the CAPS marker system consists of PCR and restriction enzyme treatment, it is much easier and less time-consuming than conventional RFLP based on Southern hybridization.

In this study, CAPS markers were developed for the analysis of genetic diversity and population structure in *A. bisporus* accessions. The newly developed CAPS markers will support the efficient evaluation and management of new and existing accessions, thereby facilitating further studies on genetic diversity and population structure in button mushrooms.

## 2. Materials and Methods

### 2.1. A. bisporus Genetic Resources and DNA Extraction

For the genetic diversity and population structure analysis, we used 41 *A. bisporus* strains provided by the Korean Mushroom Culture Collection (KMCC) at the National Institute of Horticultural and Herbal Science (NIHHS) of Rural Development Administration (RDA) in Eumseong, Republic of Korea. Geographic origins and KMCC numbers of 41 *A. bisporus* strains are given in Table 1.

*A. bisporus* strains were plated on compost dextrose agar (CDA) medium (Kisanbio, Seoul, Korea), incubated at 25 °C in the dark for 60 days, and lyophilized. DNA extraction was performed using a Plant SV mini kit (GeneAll, Seoul, Korea) according to the manufacturer protocol. Extracted DNA concentrations were quantified using an Epoch microplate spectrophotometer (BioTek, Winooski, VT, USA).

### 2.2. Primer Construction and PCR

To develop CAP markers, SNP mining was performed based on the *A. bisporus* genome and resequencing data derived from our previous study [10]. Software (dCAPS Finder 2.0) [25] was used to identify restriction enzyme recognition sites based on the resulting SNPs, and a set of 70 CAPS markers was produced (Table 2). PCR reactions (20 µL) contained 10 µL of Excel TB 2× Taq premix (Inclone Biotech, Yongin, Korea), 2 µL of 10 pmol primer (forward/reverse), 5 µL of distilled water, and 3 µL of DNA. DNA amplification was performed as follows: initial denaturation at 95 °C for 2 min; followed by 30 cycles of denaturation at 95 °C for 20 s, annealing at 55 °C for 40 s, and extension at 72 °C for 45 s; and a final extension at 72 °C for 10 min. Amplified PCR products were digested using 38 restriction enzymes. Restriction reactions (10 µL) contained 1 unit of restriction enzyme, 1 µL 10 × NEBuffer™, and 5 µL PCR product. Reactions were incubated at an appropriate temperature (Table 2) for 60 min. Digested PCR products were analyzed using 2.5% agarose gel electrophoresis.

### 2.3. Data Analysis

Using the Power Marker version 3.25 software package [26], the diversity of each CAPS marker was analyzed on the basis of five statistical parameters including major allele frequency (MAF), number of genotypes (NG), number of alleles (NA), genetic diversity (GD), and heterozygosity (He). Genetic distance was calculated using ‘’Nei’s standard’’ [27] followed by phylogeny reconstruction using rooted unweighted pair group method with arithmetic mean (UPGMA) as implemented in MEGA version 7 [28].

To visualize the relationship between the sample genotypes among the 41 *A. bisporus* accessions, principle coordinate analysis (PCoA) was conducted using GenALEx 6.5 software [29]. It was chosen to complement the UPGMA cluster analysis.

A non-hierarchical analysis of molecular variance (AMOVA) (1000 permutations) based on the degree of genetic divergence among populations was performed using GenALEx 6.5 software [29]. Population structure was analyzed based on Bayesian clustering using STRUCTURE 2.3.1 [30]. The populations number (K) was set from 1 to 10, and the populations set as location priors (LOCPRIOR) [31] under the admixture model were used to run the Markov chain Monte Carlo (MCMC) simulation algorithm. The length of the burn-in period was set to 10,000 iterations. The Delta K value was obtained by the method of Evanno [30]. The 10 runs for the optimal delta K were averaged by using the programs STRUCTURE HARVESTER [32]. Next, a hierarchical AMOVA, which was calculated considering the main groups obtained from the STRUCTURE analysis, was implemented by the software GenALEx 6.5 [29]. The statistical significance was also tested using a nonparametric approach described in Excoffier et al. (1992) with 1000 permutations [33].

The minimum number of marker sets (*n*) needed to distinguish each accession was determined using the accumulation curve approach of the “poppr package” in R [34]. For minimum marker set combinations, initial markers were selected with high GD, NA, and NG values. From the second to the last marker, markers were sequentially selected based on their ability to subdivide the highest number of accessions according to the genotyping data. Grouping based on genotyping data was performed in Microsoft Excel. After selection of minimum marker set combinations, phylogenetic analysis was used to confirm whether the accession could be fully distinguished using the minimum marker (Figure 1).

## 3. Results

### 3.1. Genotyping and Marker Diversity

Seventy CAPS markers were developed and used to assess 41 *A. bisporus* accessions. Ts polymorphisms were more common than Tv polymorphisms. The most common Ts difference was G→A, which occurred 14 times, followed by 11 C→T, 10 A→G, and 9 T→C. The most common Tv differences were C→A and C→G (four instances), followed by three A→C, A→T, G→C, G→T, T→A, and T→G changes (Table 2). The diversity index of each marker is show in Table 3. Of the 70 CAPS markers, 64 were polymorphic and 6 were monomorphic (MAF = 1). Excluding the six monomorphic markers, MAF ranged from 0.5 (AB-gCAPS-036) to 0.984 (AB-gCAPS-093), with an average of 0.698. Two NAs and two (*n* = 28) and three (*n* = 36) NGs were identified with the polymorphic markers. Similarly, excluding the six markers with one allele, He ranged from 0 (12 accessions) to 0.826 (AB-gCAPS-003), with an average of 0.290. GD ranged from 0.031 (AB-gCAPS-093) to 0.05 (AB-gCAPS-036), with an average of 0.394.

To assess whether the CAPS markers developed in this study were suitable for evaluating diversity and population structures, diversity was determined using the SSR diversity index, a widely used metric in population genetic studies [9,10,11,35]. GD, a representative diversity index, is influenced by the allele frequency. However, as SSR and SNP markers have multiple and single target locus characteristics, respectively, allele frequencies tend to differ and theoretically calculated GD index values also differ. As it is difficult to compare diversity between SNP and SSR markers using the GD index, an appropriate alternative comparison based on scaling to the maximum GD index value of each marker was used, with the following equation: $\mathrm{GD}=\left(1-\overset{k}{\underset{u=1}{\sum }}P_{lu}^{2}\right)$. When the SNP marker had a maximum of three alleles, the maximum GD value was 0.66, and the SSR marker reached a maximum GD value of 0.99 as the number of alleles increased. The average GD value of the CAPS markers in this study was 0.3599, and the minimum and maximum GD values of polymorphic markers were 0.031 and 0.5, respectively. When compared with SSR values, the average corrected SSR value, based on the maximum, was 0.540, and the minimum and maximum GD values were 0.046 and 0.750, respectively. Polymorphism frequencies were lower than average GD values of 0.548, 0.619, and 0.6807 from previously developed SSR markers. However, polymorphism levels were higher than the average GD value of 0.395 from monospores of limited accessions [9,10,11,35].

### 3.2. Grouping Based on Data Analysis and AMOVA

Multivariate and population structure analyses were performed to understand accession and population characteristics. Multivariate analysis included phylogenetic cluster analysis and principal coordinates analysis (PCoA), and population structure analysis was performed using a model-based structure. Phylogenetic tree analysis produced three groups: Group 1 (CHN 2, DEU 1, JPN 1, KOR 7, NLD 3, and USA 2), Group 2 (DEU 2, JPN 1, KOR 3, NLD 2, and USA 3), and Group 3 (CHN 3, DEU 2, JPN 4, KOR 3, NLD 1, and USA 1) (Figure 2). PCoA analysis also revealed three groups: P1 (CHN 2, DEU 1, JPN 1, KOR 7, NLD 3, and USA 2), P2 (DEU 2, JPN 1, KOR 3, NLD 1, and USA 2), and P3 (CHN 3, DEU 2, JPN 4, KOR 3, NLD 1, and USA 1) (Figure 3).

Model-based structure analysis produced two groups: POP 1 (CHN 2, DEU 1, JPN 1, KOR 7, NLD 3, and USA 2) and POP 2 (CHN 3, DEU 2, JPN 4, KOR 3, NLD 1, and USA 1), and the remaining accessions were classified as an Admix group. Population structure was revealed by classification of accessions into each group using an unrooted tree (Figure 4). Groupings were largely consistent across the three methods: Group 1, P3, and POP 1; and Group 3, P1, and POP 2 had the same accessions. Finally, Group 2 and Admix had the same accessions, with P2 having all except two of the same accessions.

To determine the degree of genetic variation and differentiation among groups, AMOVA was performed with two group types as follows: distance-based groups (Groups 1, 2, and 3) and model-based groups (POP 1, POP 2, and Admix). The variation in the population level of the two groups was 13% in the distance-based groups and 15% in the model-based groups, and the variation on an individual level was 31% and 30% among individuals in distance-based and model groups, respectively, and 55% within individuals in both groups. The *F*-statistic (*F*_ST_) value was 0.134 in the distance-based group and 0.153 in the model-based group (Table 4).

### 3.3. Selection of Minimum Markers for Discrimination

A minimal marker set for accession discrimination was developed using an accumulation curve (Figure 5A) according to the pipeline shown in Figure 1. AB-gCAPS-059 was selected as the first marker. From the second marker onwards, the phylogenetic tree was used to select markers that provided discrimination of the largest numbers of accessions. The 10 markers that were selected (AB-gCAPS-017, AB-gCAPS-022, AB-gCAPS-026, AB-gCAPS-033, AB-gCAPS-038, AB-gCAPS-039, AB-gCAPS-042, AB-gCAPS-059, AB-gCAPS-061, and AB-gCAPS-066) were able to distinguish among the 41 *A. bisporus* accessions, as confirmed using a phylogenetic tree (Figure 5B).

## 4. Discussion

### 4.1. Polymorphism Did Not Differ According to SNP Mutation Type

SNPs occur throughout the genome, with differing effects depending on the polymorphism type and location. SNPs can affect protein amino acid sequences if they occur within a coding region and introduce a codon change (non-synonymous change). Two types of SNP are found: Ts and Tv. SNP differences within the purines (A, G) or pyrimidine (C, T) nucleotides are Ts SNPs, and those that change from purine to pyrimidine or vice versa are Tv SNPs. [17]. Although there are four possible Ts changes and eight possible Tv changes, Ts SNPs occur at higher frequency than Tv SNPs [36]. Research suggests this is due to the higher number of possibly damaging non-synonymous changes resulting from Tv mutations compared with Ts mutations. Thus, Tv changes have a greater physicochemical impact on amino acid sequences and are not favored during natural selection [37]. The tv is considered to be a more drastic change than a ts, because substitution of one-ring to two-ring chemical structure or vice versa (Tv) requires more energy than substitution without change in the ring structure (Ts) [38]. The Ts/Tv ratio has been used as an important parameter in bacteria studies such as phylogenetic tree reconstruction and estimation of divergence [39].

Numbers of Ts and Tv polymorphisms were compared in the *A. bisporus* CAPS markers developed in this study. Of the 70 markers, 44 markers were Ts SNPs, and 26 markers were SNPs, a ratio of 1.7:1 for Ts:Tv. Average GD values were 0.363 (Ts) and 0.354 (Tv), showing no substantial differences. Thus, it can be inferred that the SNP type (Ts or Tv) in the *A. bisporus* marker set was not biased.

### 4.2. Diversity of Developed SNP Markers

Assessments based on molecular markers can be divided into population genetic assessments and trait assessments. SSR markers, with relatively more alleles, show a high level of diversity and are frequently used for population genetics, whereas SNP markers are frequently used to identify specific traits or determine lineage and population [40,41].

SSR and SNP markers exhibit different polymorphism traits, such as the number of repeats in the sequence, mutations of a single nucleotide, and genome-wide mutations; and the ability to obtain different types of information in the same study may facilitate the combined use of the two marker types [41,42]. SSRs were found to be better suited for detecting structure in populations at a small spatial scale with a systematic and continuous sampling design. SNP markers rather reflect ancient divergence of distant and naturally separated populations, being less sensitive to sampling design [43]. Despite these differences, both marker types were suitable for detecting the genetic structure of the fungal populations considered.

Previous studies of mushroom diversity and population structures using SSR and SNP markers focused on differences in the diversity index for the same marker [9,10,11,12,13,14,44]. However, direct comparison of diversity was difficult because of differences in the target loci of the two marker types. To solve these problems, the diversity index of the two markers was calculated and compared using an equation based on the maximum diversity index of each marker. This comparison method can be applied to a range of markers to facilitate comparative analysis of mushroom diversity and population structure.

### 4.3. Population Structure Analysis Using SNP Markers

In previous studies, SNP markers showed a similar grouping pattern and more accurate lineage classification of the neighbor joining tree in population structure analysis compared to SSR markers [41,42]. In particular, the explanatory value for the first main coordinate in PCoA analysis was higher for SNP markers, and this was a common phenomenon regardless of accession [41,42]. No common characteristics were observed in the clustered accessions of all groups with the three clustering methods used in this study, and collection location did not correlate with accessions in the clusters. This may be because the SNP markers classified lineage, and the current commercial accessions were divided into relatively few lineages. In addition, accessions in geographically separated regions had similar sequences, consistent with the genetically similar nature of accessions cultivated worldwide [4].

Population structure analysis is used to identify characteristics by composing clusters according to accession similarities. This led us to speculate that similarities among sequences would have a strong impact on accession clustering, and we therefore used AMOVA to confirm the extent to which genetic variation between populations and accessions were affected by one another. AMOVA of a model-based population revealed slight difference between groups and accessions by collection area (13% and 86%, respectively) and genotype (15% and 85%, respectively). Population variation according to genotype and collection area did not differ substantially, consistent with most commercially cultivated button mushrooms being derived from similar strains.

### 4.4. Selection of Minimum Markers for Accession Identification

Development of a minimum set of molecular markers to distinguish accessions provides the basis for future evaluation of existing new *A. bisporus* resources. An accumulation curve, in which SNP loci from the 70 CAPS markers were randomly identified and calculated [34], was used to determine the minimum number of loci required to distinguish between accessions. This approach can be extended to determine minimum marker numbers for future applications. In this study, 10 SNP markers were sufficient to distinguish 41 *A. bisporus* accessions. Previous research also developed minimum marker sets for accession differentiation: SSR markers were used to distinguish all accessions by using 4 of 26 accessions [10] and 6 of 171 accessions [11]. Discrimination using SNP markers tends to require more markers, and distinguish fewer accessions, than SSR markers. SNP markers may therefore be less efficient when examining large numbers of accessions; however, linkages determined using SNP markers may be more stable than linkages established using SSR markers.

Currently, studies using SNP markers in button mushrooms focus on the evaluation of accessions by direct comparison of SNPs by sequencing or through QTL mapping related to characteristics such as mushroom color and robustness [13,14,15,16]. Advances in genomic analysis have facilitated the efficient use of SSR and SNP markers for population genetic studies of accession collections. However, studies of genetic diversity and population structure using CAPS markers in mushrooms are limited. The SNP markers developed in this study will facilitate the comparison and evaluation of existing *A. bisporus* accessions and will provide the basis of future analysis and management of new accessions.

## 5. Conclusions

Consumer demand for new button mushroom varieties has increased alongside the recent growth of the mushroom industry. Optimal selection of breeding materials through the evaluation and management of accession collections is important for the development of new varieties. Several molecular markers have been used to evaluate crops with restrictive genotypes, such as button mushrooms. Of these, SNP markers are widely used for association mapping, accession discrimination, and analysis of genetic diversity and population structure. However, only limited numbers of molecular markers are available to support *A. bisporus* breeding strategies. In this study, a set of 70 CAPS markers was developed to analyze the diversity and population structure of 41 *A. bisporus* accessions. Of the 70 markers, a set of 10 minimum markers was identified that was able to identify all 41 *A. bisporus* accessions. The developed CAPS markers will be useful for analysis of button mushroom diversity and population structures, and will also be useful for variety identification.

## Figures and Tables

**Figure 1 jof-07-00375-f001:**
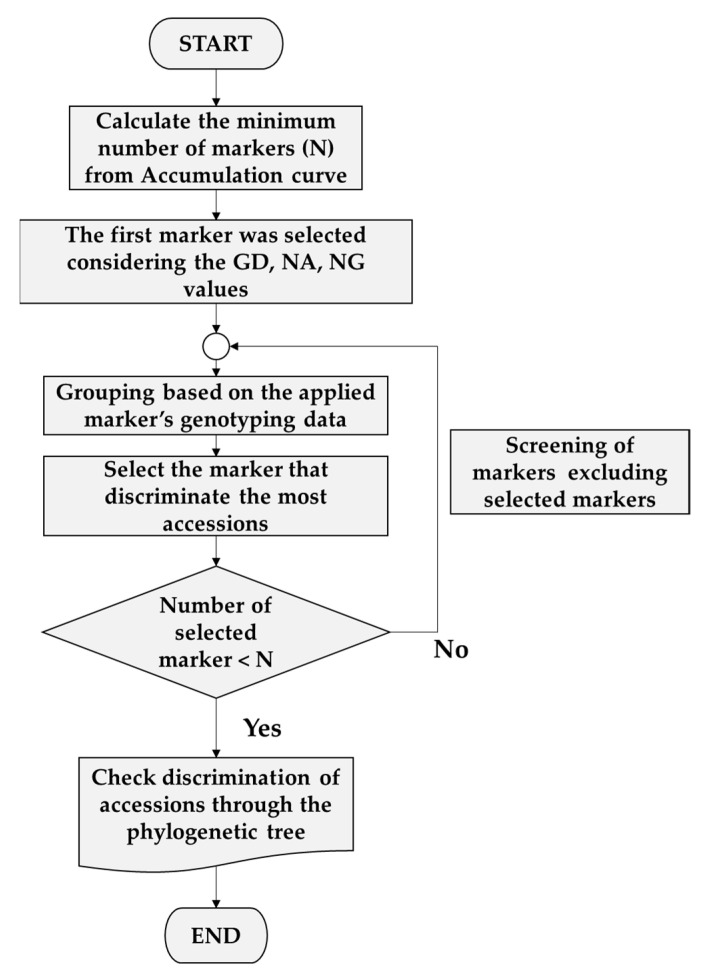
Pipeline used for minimum marker set selection.

**Figure 2 jof-07-00375-f002:**
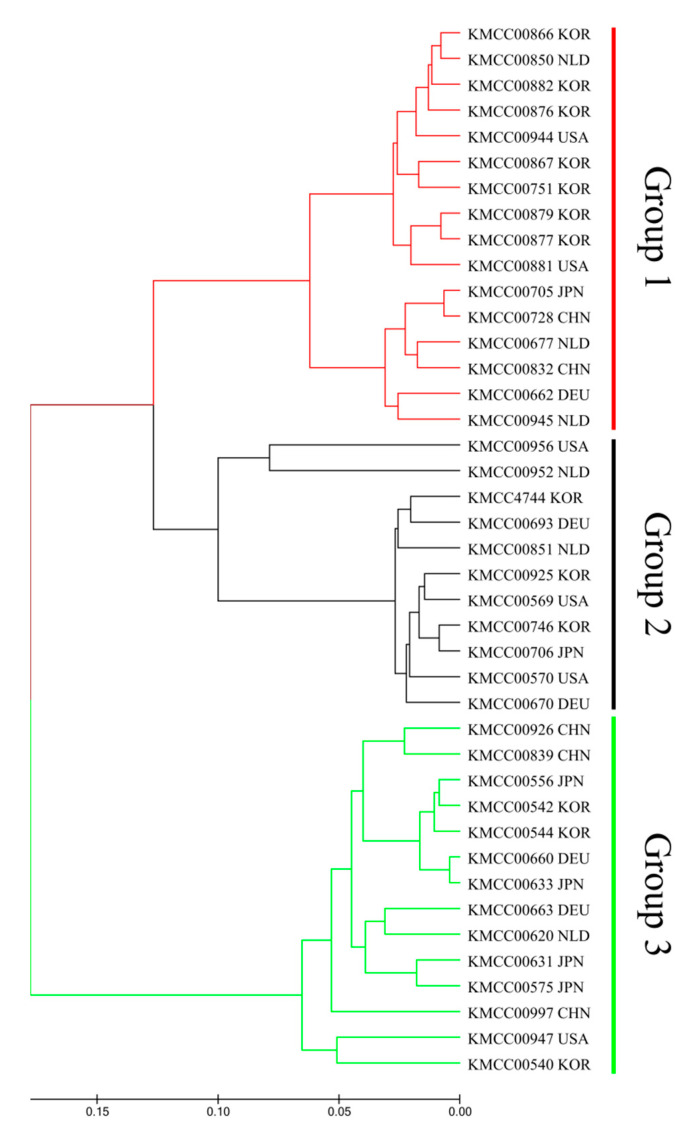
Phylogenetic tree constructed using the unweighted pair pop method with arithmetic mean based on Nei’s genetic distance.

**Figure 3 jof-07-00375-f003:**
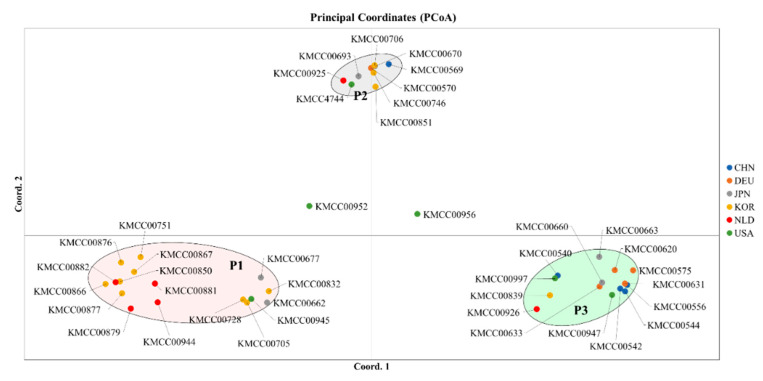
Principal coordinates analysis to identify the characteristics and members of each cluster. Accessions were divided into three main groups, with two central accessions not classified.

**Figure 4 jof-07-00375-f004:**
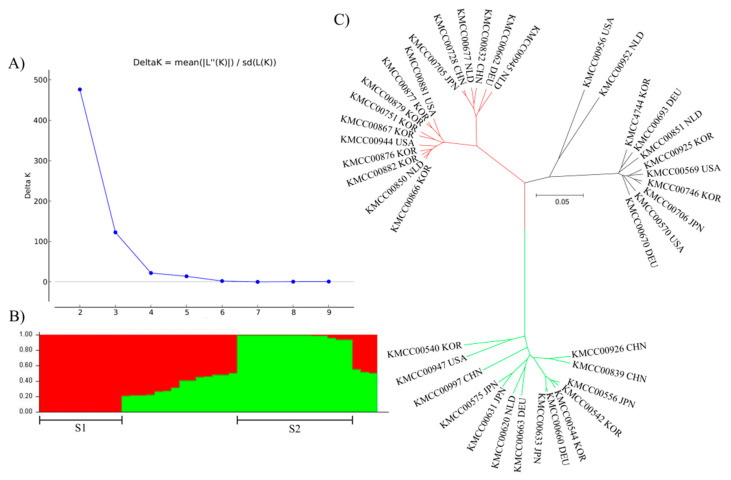
Group structure analysis of *Agaricus bisporus* accessions. The most appropriate number of groups was determined by repeating five iterations using Structure Burn-in and Markov Chain Monte Carlo (MCMC) with 100,000 iterations. (**A**) Delta K values were calculated using Structure Harvester. (**B**) Probability of each accession belonging to a group. (**C**) Unrooted tree confirmation of population structure.

**Figure 5 jof-07-00375-f005:**
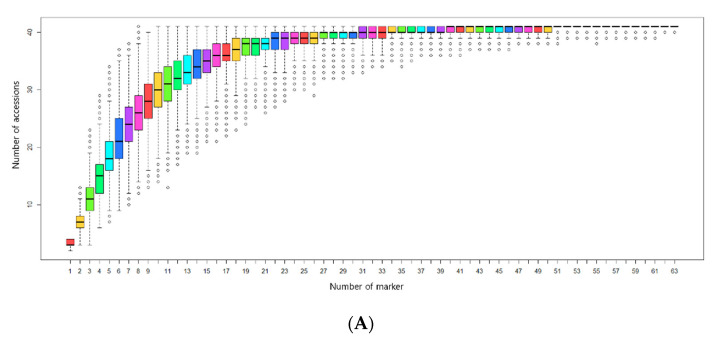
(**A**) The number of markers sufficient to serve as a minimum marker was determined using an accumulation curve calculated using the poppr package in R. (**B**) Phylogenetic tree illustrating discrimination of 41 *Agaricus bisporus* accessions using 10 selected markers.

**Table 1 jof-07-00375-t001:** *A. bisporus* accessions used for validation of CAPS markers.

Strain Number	Geographic Region	Strain Number	Geographic Region
KMCC00540	KOR	KMCC00832	CHN
KMCC00542	KOR	KMCC00839	CHN
KMCC00544	KOR	KMCC00850	NLD
KMCC00556	JPN	KMCC00851	NLD
KMCC00569	USA	KMCC00866	KOR
KMCC00570	USA	KMCC00867	KOR
KMCC00575	JPN	KMCC00876	KOR
KMCC00620	NLD	KMCC00877	KOR
KMCC00631	JPN	KMCC00879	KOR
KMCC00633	JPN	KMCC00881	USA
KMCC00660	DEU	KMCC00882	KOR
KMCC00662	DEU	KMCC00925	KOR
KMCC00663	DEU	KMCC00926	CHN
KMCC00670	DEU	KMCC00944	USA
KMCC00677	NLD	KMCC00945	NLD
KMCC00693	DEU	KMCC00947	USA
KMCC00705	JPN	KMCC00952	NLD
KMCC00706	JPN	KMCC00956	USA
KMCC00728	CHN	KMCC00997	CHN
KMCC00746	KOR	KMCC4744	KOR
KMCC00751	KOR		

**Table 2 jof-07-00375-t002:** CAPS markers developed using SNPs mined from *A. bisporus* resequencing data.

CAPS Marker	SNP Locus	Chr.	Substitution	Ref	Alt	Restriction Enzyme	Temp. (°C)	Left (L)/Right (R) Primer
AB-gCAPS-001	HD1 homeodomain transcription factor A mating type protein	Chr01	Transversion	C	A	Fnu4HI	37	L–TGTCAATGTCAATTCAGACTCC
								R–TTTGAATATCGTGTTGCAGAGA
AB-gCAPS-002	HD1 homeodomain transcription factor A mating type protein	Chr01	Transition	T	C	NdeI	37	L–CTCAATCGAGGACGAGTTATTC
								R–GTTCGAGCTCAAAATCAAGTTC
AB-gCAPS-003	HD1 homeodomain transcription factor A mating type protein	Chr01	Transversion	G	T	HpyCH4V	37	L–CTCAATCGAGGACGAGTTATTC
								R–GTTCGAGCTCAAAATCAAGTTC
AB-gCAPS-004	PIF1 protein	Chr01	Transition	A	G	BtsCI	50	L–ACTACAGTATCCACCAATTGCC
								R–ATAGGGATTTAGTTGCCATGTG
AB-gCAPS-005	PIF1 protein	Chr01	Transition	G	A	BsaBI	60	L–TGATTCTTCCAAAGTTTGAGGT
								R–ATGGCCTCTTATACTGGTGTTG
AB-gCAPS-006	Transposon Tf2-11 polyprotein	Chr02	Transversion	A	C	BtsCI	50	L–GAACCGTCTTGGTACTATTTGC
								R–TAAGGCAGAACGTCTAGAGGAA
AB-gCAPS-007	Transposon Tf2-11 polyprotein	Chr02	Transition	A	G	MseI	37	L–CCACACTCCTCGCATCTATATT
								R–GTGGGTACAAAGACAAAGGAAA
AB-gCAPS-008	Oleate activated transcription factor 3, partial	Chr06	Transition	T	C	TaqI-v2	65	L–CTCAGCCATCTCTACCTCTCTC
								R–ACATGTACAAGACCGTCAATCA
AB-gCAPS-009	ATP20 subunit G of the mitochondrial F1F0 ATP synthase	Chr07	Transversion	G	C	Hpy99I	37	L–TTAGGTGTACAAAGACAATGCG
								R–CTTCTCCAACTCTTTAACGCTC
AB-gCAPS-012	ATP20 subunit G of the mitochondrial F1F0 ATP synthase	Chr07	Transversion	C	A	HpyCH4IV	37	L–CATCATCTGTTGTGGTCATCTC
								R–AGTGAGGCAATAAAATGGAAGA
AB-gCAPS-013	ATP20 subunit G of the mitochondrial F1F0 ATP synthase	Chr07	Transition	C	T	HaeIII	37	L–CATCATCTGTTGTGGTCATCTC
								R–AGTGAGGCAATAAAATGGAAGA
AB-gCAPS-015	Polyphenol oxidase	Chr08	Transition	G	A	TaqI-v2	65	L–GACCGTCAATTCTCTCTTTACG
								R–AATCAAAACATAAGGACGATGC
AB-gCAPS-016	Polyphenol oxidase	Chr08	Transversion	A	C	MseI	37	L–AAGTCATCTCCCTACCCAAAGT
								R–TCGACTTTTATCAGACCCATTT
AB-gCAPS-017	WC-1 blue light photoreceptor	Chr08	Transition	G	A	BtsCI	50	L–GTTCTGGAAGTAAAGCGAAGAC
								R–CGTAGAACACAAAGTCTTGCAG
AB-gCAPS-018	Serine/threonine-protein kinase ATG1	Chr09	Transition	A	G	HpyAV	37	L–GCAAGACTAGAGGGTGATGAAG
								R–TATGTCCTTGTGGACGATACAA
AB-gCAPS-019	Serine/threonine-protein kinase ATG1	Chr09	Transversion	A	C	ApoI	50	L–CCTCCGATTGTTATCCATAGTC
								R–GAGGTGTTAGATCCCAAAGCTA
AB-gCAPS-020	Serine/threonine-protein kinase Ppk19	Chr09	Transition	G	A	NcoI	37	L–ACCGTCTATCCCACAATGTTAG
								R–GAATATGTTACCAGCATGGTCC
AB-gCAPS-021	Serine/threonine-protein kinase Ppk19	Chr09	Transversion	A	T	BtsCI	50	L–TAGCAGTCTCTATGCTGGACAA
								R–CCGGATACAGGAAGAACACTTA
AB-gCAPS-022	Cytochrome P450	Chr12	Transversion	T	A	HinfI	37	L–CATCGAAGCTGATGAGTACAAC
								R–CGAATAGAACTGTCGAGTTTCC
AB-gCAPS-024	Transposon Tf2-11 polyprotein	Chr12	Transversion	T	G	BccI	37	L–GTCTCCATTCTTCTAAACACCG
								R–AGCACCAGAACTGGATAAAGAA
AB-gCAPS-025	Retrovirus-related Pol polyprotein	Chr13	Transition	G	A	Hpy99I	37	L–CAAGTATCAAATGGAACTGCCT
								R–AGATATACACACCGAAGGATGG
AB-gCAPS-026	Retrovirus-related Pol polyprotein	Chr13	Transition	C	T	DpnI	37	L–CAAATCTGCCTTCGTATTCATT
								R–ATCATCAATCACGTCGAACATA
AB-gCAPS-028	hypothetical protein AGABI2DRAFT_123095	Chr02	Transition	T	C	HpyCH4V	37	L–GTTTATCAAGTTCATCAAGCCC
								R–TCTGGGTGCTTCTGTATTCTTT
AB-gCAPS-030	hypothetical protein AGABI2DRAFT_196053	Chr02	Transition	C	T	RsaI	37	L–GAACCAATTCACAGTGGTTTCT
								R–TACTTTATGGAGCCGTCAGAAT
AB-gCAPS-031	hypothetical protein AGABI2DRAFT_146035	Chr02	Transition	C	T	Hpy99I	37	L–AACGTCACTCTTACTCATGCAA
								R–TACATGAATGCCTCATGTTGTT
AB-gCAPS-032	hypothetical protein AGABI2DRAFT_146035	Chr02	Transversion	T	G	MnlI	37	L–ATTACTGGGATGATGACTCTCG
								R–AGGAGGGGTAGGGACTCTG
AB-gCAPS-033	hypothetical protein AGABI2DRAFT_193507	Chr03	Transversion	C	A	BsmI	65	L–ATGTTGACATGTTGGACAGAAA
								R–ATGGTGCCGTTGTAGTCTTACT
AB-gCAPS-034	hypothetical protein AGABI2DRAFT_193507	Chr03	Transition	A	G	Hpy166II	37	L–AGGGACGCTTAAAATTACCTGT
								R–GTCTCCAAACTCGTCAGTTCTC
AB-gCAPS-035	hypothetical protein AGABI2DRAFT_193507	Chr03	Transversion	C	G	MspI	37	L–GCCATATGTCACCTCAGAAAAT
								R–TTCTTTTTCTCAGGATGTTGCT
AB-gCAPS-036	hypothetical protein AGABI1DRAFT_131635	Chr03	Transition	G	A	PvuI	37	L–CAATGGTGATCTAAGCACTCAA
								R–AGGGAGACGAAGACAAAACATA
AB-gCAPS-037	hypothetical protein AGABI2DRAFT_176555	Chr03	Transition	C	T	HaeIII	37	L–GTGCTTATCCTCGAATGTCTTC
								R–GAATCGTCGGGATATAATGTTG
AB-gCAPS-038	hypothetical protein AGABI2DRAFT_141360	Chr03	Transition	G	A	Hpy188I	37	L–GTCACAGCAGCAAAGAAATACA
								R–GGTGAGATTAGACAGAGGTTCG
AB-gCAPS-039	hypothetical protein AGABI2DRAFT_200532	Chr03	Transversion	T	A	BtsCI	50	L–TGAGGATAGCGAAAGAAGAGAG
								R–TAGCCTTCGATTTAAGTTCAGC
AB-gCAPS-041	hypothetical protein AGABI2DRAFT_71082	Chr03	Transition	C	T	HaeIII	37	L–TATCTTGGTATTTACGATGGCG
								R–TGTACTCAGCAGTCTTGTGCTC
AB-gCAPS-042	hypothetical protein AGABI2DRAFT_71082	Chr03	Transition	T	C	NlaIV	37	L–ATAACAATGGCCATCAAACTCT
								R–CGGCTCTCGTATAGAATGAATC
AB-gCAPS-043	hypothetical protein AGABI2DRAFT_71082	Chr03	Transition	A	G	BsrDI	65	L–GATGCCACTTTAGACTTTTTGG
								R–TATGGTAAATGGAAAAGATCCG
AB-gCAPS-045	hypothetical protein AGABI2DRAFT_121386	Chr04	Transition	A	G	HpyCH4V	37	L–AACAGACTGACCTCACAAAACC
								R–CGTCGTTATCTTTCCATTTGAT
AB-gCAPS-047	hypothetical protein AGABI1DRAFT_90407	Chr04	Transition	G	A	HpyCH4V	37	L–CTCTTAGCGAGGCGTTATCTTA
								R–ATTGGAACATAATTCATTGGGA
AB-gCAPS-048	hypothetical protein AGABI2DRAFT_117363	Chr04	Transition	G	A	Hpy188I	37	L–TTCCTTAAGCCAGTTTTGAAGA
								R–GAGGATTGGACTAATACCGTGA
AB-gCAPS-050	hypothetical protein AGABI2DRAFT_178864	Chr06	Transition	A	G	HphI	37	L–CACTACTTCCCCTCCTCTCTTT
								R–ACTACCAAAAGAGGCATCTCAA
AB-gCAPS-051	hypothetical protein AGABI2DRAFT_119143	Chr06	Transition	G	A	HaeIII	37	L–AGTTAGCTATTGCCTGAGCTTG
								R–GTCACAAGCCATCTCAATCTTT
AB-gCAPS-052	hypothetical protein AGABI2DRAFT_119143	Chr06	Transition	T	C	HpyCH4III	37	L–GGTTTTCTAGTGCCGTAGTGAG
								R–TTCTCAATGACCCTTTGAACTT
AB-gCAPS-053	hypothetical protein AGABI2DRAFT_119143	Chr06	Transversion	A	T	MluCI	37	L–TCAGTAAACTCCCTACGCTCAT
								R–GCCTAGCCGTAAGTTCACATAA
AB-gCAPS-054	hypothetical protein AGABI1DRAFT_126593	Chr06	Transition	A	G	SfaNI	37	L–CAACAATGTCTCCTTGAGTCCT
								R–TTTCAGTTTGCATTCTCTGATG
AB-gCAPS-055	hypothetical protein AGABI1DRAFT_126593	Chr06	Transversion	A	T	ApoI	50	L–GATCCCCAAATAATGAATGCTA
								R–TATACTCCCGACGTAGAACAGC
AB-gCAPS-056	hypothetical protein AGABI1DRAFT_126595	Chr06	Transversion	T	G	BtsCI	50	L–GATGGTCACGATTTGTTTCTTT
								R–AACAAACCTCATTATTTCTGCC
AB-gCAPS-058	hypothetical protein AGABI2DRAFT_179115, partial	Chr06	Transversion	C	G	RsaI	37	L–GTTTCTGGAGGGAGTATACGTG
								R–ATCACATGTCAAGTTGTGGAGA
AB-gCAPS-059	hypothetical protein AGABI2DRAFT_225478	Chr10	Transition	C	T	Tsp45I	65	L–CAGTGGTACGACGTTCAAAATA
								R–ACACCAATTATGGTCTCGATTC
AB-gCAPS-061	hypothetical protein AGABI1DRAFT_133092, partial	Chr10	Transition	G	A	HphI	37	L–ACAAAACGAGAAGAGCAGAGAG
								R–CTAATACGATTTACGATGGCGT
AB-gCAPS-062	hypothetical protein AGABI1DRAFT_133088	Chr10	Transition	G	A	BssSI	37	L–CTCGAGATAGCAGAGGAGCAT
								R–TACAACGCATCGTACTCAAAAC
AB-gCAPS-063	hypothetical protein AGABI1DRAFT_133088	Chr10	Transversion	G	T	BssSI	37	L–AGCTTTTGCACGAGATGAATAC
								R–AGGAAGGTTGAGAAAGGGATAG
AB-gCAPS-064	hypothetical protein AGABI2DRAFT_194394	Chr10	Transversion	C	G	BstAPI	60	L–GTCTCTTCATCGAAACCATCTC
								R–TTTGGCATCATTCATTACTTCA
AB-gCAPS-065	hypothetical protein AGABI2DRAFT_179918	Chr10	Transition	C	T	BstNI	60	L–CCTTATTCTTGTGATTGAAGGC
								R–GACATTTGGTGCAGGAGTAGAT
AB-gCAPS-066	hypothetical protein AGABI2DRAFT_74687	Chr10	Transversion	G	C	XcmI	37	L–AAGTCCGCAATTGACCTACTAA
								R–AGTGTGCAAAATTGAGGAGAGT
AB-gCAPS-068	hypothetical protein AGABI2DRAFT_74687	Chr10	Transversion	G	C	HpyCH4III	37	L–GACGTCCAAAATCTTGAGTGAT
								R–GACGTTGGTCTCAGCTTACTTC
AB-gCAPS-070	hypothetical protein AGABI1DRAFT_48245, partial	Chr10	Transition	T	C	MluCI	37	L–CTTCGGAAATATGTCTTCAAGG
								R–GCGAGGTATCAGAGGAATGTAG
AB-gCAPS-071	hypothetical protein AGABI1DRAFT_48245, partial	Chr10	Transition	A	G	Hpy188I	37	L–AACCTCATTCCCAACCTTATCT
								R–AATATATTGGTCATTGGAACCG
AB-gCAPS-072	hypothetical protein AGABI1DRAFT_48245, partial	Chr10	Transition	G	A	BstNI	60	L–TTGTAGCTTATGACATGGTTCG
								R–GGAATTATTTTGACGGTTTGAA
AB-gCAPS-073	Uncharacterized protein Hypma_04748, partial	Chr10	Transition	C	T	TaqI	65	L–TATTGATCTCAGCCAACCTTTT
								R–TCCTCACTTTTAGGAGGATCAA
AB-gCAPS-078	hypothetical protein AGABI2DRAFT_195493	Chr11	Transition	C	T	Hpy166II	37	L–CTAGGATCATATGCGATTTTGC
								R–ATAGAACTCAACGCCGACAG
AB-gCAPS-081	hypothetical protein AGABI2DRAFT_68830	Chr11	Transition	T	C	BsmI	65	L–ATTTTTCAGGTCACGTTCTCAC
								R–TAGATGGTTAAACGTGTGGGAT
AB-gCAPS-082	hypothetical protein AGABI2DRAFT_68830	Chr11	Transition	A	G	HinfI	37	L–GTAAAAACAGTTTCCGAAGCAC
								R–TATTTCTCAACAGGAGTGACCC
AB-gCAPS-083	hypothetical protein AGABI2DRAFT_196017, partial	Chr11	Transition	C	T	MnlI	37	L–GATCTATACTTCGGCGATTGAG
								R–ACTATAGAGAGTGCCACCAGGA
AB-gCAPS-084	hypothetical protein AGABI2DRAFT_229511	Chr11	Transition	G	A	BtsI	55	L–TGGAATTAATAAGGCATTTTGG
								R–ATCGACCTCTGATATTCACGAT
AB-gCAPS-086	hypothetical protein AGABI2DRAFT_79146	Chr11	Transversion	T	A	HphI	37	L–CCCAATTCCTATCATGCTTATC
								R–ATACTGACCATCGCCACTATGT
AB-gCAPS-087	hypothetical protein AGABI1DRAFT_77545	Chr11	Transition	T	C	SfaNI	37	L–TGCAATCGCTTTGTAAGTATCA
								R–ATCCCTATACCCATCGCTAGTT
AB-gCAPS-088	hypothetical protein AGABI1DRAFT_77545	Chr11	Transversion	G	T	BbsI	37	L–AATCATTCGACCAATGCTAATC
								R–ACCATCCTGACCACTCTATTTG
AB-gCAPS-089	hypothetical protein AGABI1DRAFT_77545	Chr11	Transversion	C	A	BstBI	65	L–TCGTACCATAGAACCCTTGACT
								R–TTGGCTTCTACAACCCTTACAT
AB-gCAPS-090	hypothetical protein AGABI1DRAFT_77545	Chr11	Transversion	C	G	HpyAV	37	L–AGAAAGGTGAAGACTCACGGTA
								R–GGGTTGTTGTTTTCAGCTTATC
AB-gCAPS-093	hypothetical protein AGABI2DRAFT_188752	Chr11	Transition	T	C	Hpy188I	37	L–AATCCTAGAATCACTTCAGCCA
								R–CACCTCATTCCGAATTATTCAT

Chr., chromosome; Ref., reference nucleotide; Alt, alternative nucleotide; Temp., restriction reaction incubation temperature.

**Table 3 jof-07-00375-t003:** Genetic diversity index of 41 *Agaricus bisporus* accessions assessed with 70 CAPS markers.

Marker	MAF ^1^	NG ^2^	NA ^3^	GD ^4^	He ^5^
AB-gCAPS-001	0.564	3	2	0.492	0.256
AB-gCAPS-002	0.609	3	2	0.476	0.696
AB-gCAPS-003	0.587	2	2	0.485	0.826
AB-gCAPS-004	1.000	1	1	0.000	0.000
AB-gCAPS-005	0.750	3	2	0.375	0.250
AB-gCAPS-006	1.000	1	1	0.000	0.000
AB-gCAPS-007	1.000	1	1	0.000	0.000
AB-gCAPS-008	0.694	2	2	0.425	0.613
AB-gCAPS-009	0.975	2	2	0.049	0.000
AB-gCAPS-012	0.700	2	2	0.420	0.000
AB-gCAPS-013	0.724	3	2	0.400	0.079
AB-gCAPS-015	0.720	3	2	0.404	0.463
AB-gCAPS-016	0.744	2	2	0.381	0.513
AB-gCAPS-017	0.738	2	2	0.387	0.525
AB-gCAPS-018	0.750	2	2	0.375	0.500
AB-gCAPS-019	0.732	3	2	0.393	0.341
AB-gCAPS-020	0.659	3	2	0.450	0.390
AB-gCAPS-021	0.756	2	2	0.369	0.488
AB-gCAPS-022	0.793	3	2	0.329	0.317
AB-gCAPS-024	0.683	2	2	0.433	0.634
AB-gCAPS-025	0.765	2	2	0.360	0.000
AB-gCAPS-026	0.770	2	2	0.354	0.459
AB-gCAPS-028	0.650	2	2	0.455	0.700
AB-gCAPS-030	0.610	2	2	0.476	0.000
AB-gCAPS-031	0.951	2	2	0.093	0.098
AB-gCAPS-032	1.000	1	1	0.000	0.000
AB-gCAPS-033	0.586	3	2	0.485	0.371
AB-gCAPS-034	0.561	3	2	0.493	0.146
AB-gCAPS-035	0.598	3	2	0.481	0.756
AB-gCAPS-036	0.500	3	2	0.500	0.294
AB-gCAPS-037	1.000	1	1	0.000	0.000
AB-gCAPS-038	0.817	2	2	0.299	0.366
AB-gCAPS-039	0.549	3	2	0.495	0.512
AB-gCAPS-041	0.537	3	2	0.497	0.585
AB-gCAPS-042	0.625	3	2	0.469	0.500
AB-gCAPS-043	0.662	3	2	0.448	0.618
AB-gCAPS-045	0.793	3	2	0.329	0.366
AB-gCAPS-047	0.855	2	2	0.248	0.289
AB-gCAPS-048	0.694	3	2	0.425	0.226
AB-gCAPS-050	0.750	3	2	0.375	0.289
AB-gCAPS-051	0.634	3	2	0.464	0.244
AB-gCAPS-052	0.615	3	2	0.473	0.205
AB-gCAPS-053	0.577	3	2	0.488	0.692
AB-gCAPS-054	0.973	2	2	0.053	0.000
AB-gCAPS-055	0.625	2	2	0.469	0.750
AB-gCAPS-056	0.771	3	2	0.353	0.057
AB-gCAPS-058	0.577	3	2	0.488	0.128
AB-gCAPS-059	0.524	3	2	0.499	0.610
AB-gCAPS-061	0.608	3	2	0.477	0.135
AB-gCAPS-062	0.671	3	2	0.442	0.171
AB-gCAPS-063	0.768	2	2	0.356	0.463
AB-gCAPS-064	0.775	2	2	0.349	0.450
AB-gCAPS-065	0.713	3	2	0.410	0.125
AB-gCAPS-066	0.632	2	2	0.465	0.000
AB-gCAPS-068	0.775	3	2	0.349	0.350
AB-gCAPS-070	0.585	2	2	0.485	0.000
AB-gCAPS-071	0.615	2	2	0.473	0.000
AB-gCAPS-072	0.600	2	2	0.480	0.000
AB-gCAPS-073	0.848	2	2	0.257	0.000
AB-gCAPS-078	0.634	3	2	0.464	0.098
AB-gCAPS-081	0.613	3	2	0.475	0.025
AB-gCAPS-082	0.667	2	2	0.444	0.000
AB-gCAPS-083	0.526	3	2	0.499	0.231
AB-gCAPS-084	1.000	1	1	0.000	0.000
AB-gCAPS-086	0.675	2	2	0.439	0.000
AB-gCAPS-087	0.671	3	2	0.442	0.073
AB-gCAPS-088	0.878	3	2	0.214	0.049
AB-gCAPS-089	0.850	3	2	0.255	0.100
AB-gCAPS-090	0.902	3	2	0.176	0.098
AB-gCAPS-093	0.984	2	2	0.031	0.031
Mean	0.7248	2.4	1.9	0.3599	0.2650

^1^ Major allele frequency ^2^ Number of genotype ^3^ Number of allele ^4^ Gene Diversity ^5^ Heterozygosity.

**Table 4 jof-07-00375-t004:** AMOVA analysis of distance- and model-based clustering.

Distance-Based Group						
Source	df	SS	MS	Est. Var.	Percentage	*F* _ST_
Among Pops	2	147.844	73.922	2.062	13%	0.134
Among Indiv	38	691.058	18.186	4.849	31%	
Within Indiv	41	348.000	8.488	8.488	55%	
Total	81	1186.902		15.399	100%	
**Model-Based Population**						
**Source**	**df**	**SS**	**MS**	**Est. Var.**	**Percentage**	***F*_ST_**
Among Pops	2	163.971	81.985	2.377	15%	0.153
Among Indiv	38	674.932	17.761	4.637	30%	
Within Indiv	41	348.000	8.488	8.488	55%	
Total	81	1186.902		15.501	100%	

## Data Availability

The data for this manuscript are available at NCBI GenBank numbers CP039873–CP039885.

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
