# Peer review of "Development of CAPS Markers for Evaluation of Genetic Diversity and Population Structure in the Germplasm of Button Mushroom (Agaricus bisporus)"

_jof, 2021, doi:10.3390/jof7050375_

Round 1

Reviewer 1 Report

Hyejin An et al., used SNP mining to develop CAPS markers to distinguish A. bisporus diversity. They identified 10 minimum markers to identify all A. bisporus diversity and population structures and will also be useful for variety identification. In general, this manuscript will provide important and useful information for comparison and evaluation of existing A. bisporus populations.

My main comments are as follow:

Experimental procedures are insufficiently presented. It can be improved by proving more information specially on the elements of statical analysis and software used.

Results, I am abit confused by accession and KMCC numbers mentioned by authors, are they the same thing, where did you get your accession from?

L230-243, this data should have been presented in the results section, not here.

Discussion session is the repeat of what is already said, if possible, provide more comparison to other published studies.

I have few comments that authors can address:

L23-26, very long sentence and vague. Re-write.

Abstract and throughout, the authors keep referring to accessions, by that do they mean genomes of A. bisporus found in GenBank?

L87, give company details of CDA.

L94, did resequencing was done for this work only? how it was performed?

L96, provide reference for the software used.

L98, PCR reaction, mention what was the other 5ul used in the PCR?

Author Response

From

Ick-Hyun Jo

Department of Herbal Crop Research, National Institute of Horticultural and Herbal Science (NIHHS), Rural Development Administration (RDA), Eumseong 27709, Republic of Korea

[Tel.: +82-43-871-5613]

[Fax: +82-43-871-5639]

E-mail: intron@korea.kr

Jong-Wook Chung

Department of Industrial Plant Science and Technology, Chungbuk National University Cheongju

28644, Republic of Korea

[Tel.: +82-43-261-2524]

[Fax: +82-43-271-0413]

Email address: jwchung73@chungbuk.ac.kr

To

The Reviewer

Journal of Fungi

Dear Sir,

Sub: Regarding submission of the revised manuscript with id- jof-1200916-reg.

Herewith I am submitting our revised manuscript entitled of

Development of CAPS markers for evaluation of genetic diversity and population structure in the germplasm of button mushroom (Agaricus bisporus) manuscript id- jof-1200916 towards the publication in your prestigious journal- Journal of Fungi. As per your direction and reviewer recommendation, I assure you that all the corrections have been carried out according to reviewer’s comments and suggestions. I am looking forward to your favorable reply with optimism.

With kind regards,

Sincerely

Ick-Hyun Jo & Jong-Wook Chung

Reviewer Comment and response

 Reviewer 1 Comment #1: Experimental procedures are insufficiently presented. It can be improved by proving more information specially on the elements of statical analysis and software used.

Responses the Reviewer 1 Comment #1: Thank you so much for your valuable suggestion.

To ensure the manuscript presents all necessary details, you requested that we provide detailed descriptions of the software and statistical analysis methods that we used in this study. We agree with you that this information is essential. Based on the reasonable and constructive reviewers’ suggestions, we have made additional modifications and highlighted with yellow color in the revised text [L121-142].

Reviewer 1 Comment #2: Results, I am abit confused by accession and KMCC numbers mentioned by authors, are they the same thing, where did you get your accession from?

Responses the Reviewer 1 Comment #2: The KMCC (Korean Mushroom Culture Collection) number is an A. bisporus genetic resource deposited in the National Institute of Horticultural and Herbal Science (NIHHS) of Rural Development Administration (RDA). We determined that accession and KMCC numbers could cause confusion, so we wrote the source of the A. bisporus germplasm provided in the materials and methods part and revised Table 1 [L93-97].

 Reviewer 1 Comment #3: L230-243, this data should have been presented in the results section, not here.

 Responses the Reviewer 1 Comment #3: Thank you for these critical comments. The pointed paragraph has been deleted from the discussion session and then moved to the results section [L168-185].

 Reviewer 1 Comment #4: Discussion session is the repeat of what is already said, if possible, provide more comparison to other published studies

Responses the Reviewer 1 Comment #4: Thank you for these critical comments. As reviewer’s comments, we revised the manuscript. changes of discussion session are highlighted with green color.

 Minor point

Reviewer 1 Comment #5: L23-26, very long sentence and vague. Re-write.

Responses the Reviewer 1 Comment #5: Based on the reasonable reviewers’ suggestions, we corrected the sentence [L23-27].

Reviewer 1 Comment #6: Abstract and throughout, the authors keep referring to accessions, by that do they mean genomes of A. bisporus found in GenBank?

Responses the Reviewer 1 Comment #6: The accession mentioned in manuscript refers to the various button mushroom strains (germplasm) collected from different countries. In general, the composition of germplasm includes newly collected resources, landrace and breeding lines, which are commonly referred to as accession. Therefore, the button mushroom strain used in this study was described as an accession.

Reviewer 1 Comment #7: L87, give company details of CDA.

Responses the Reviewer 1 Comment #7: As reviewer’s comments, we provided the source of the CDA medium [L98-99]

 Reviewer 1 Comment #8: L94, did resequencing was done for this work only? how it was performed?

Responses the Reviewer 1 Comment #8: The A. bisporus resequencing data used in this study was not newly analyzed, but derived from our previous study. [Previous study: Development of Polymorphic Simple Sequence Repeat Markers using High-Throughput Sequencing in Button Mushroom (Agaricus bisporus)]. Therefore, to prevent confusion, the source of the resequencing data used for SNP mining was additionally written and reflected in the manuscript [L105-106].

Reviewer 1 Comment #9: L96, provide reference for the software used.

Responses the Reviewer 1 Comment #9: As reviewer’s comments, we have added a reference to the software used [L107].

Reviewer 1 Comment #10: L98, PCR reaction, mention what was the other 5ul used in the PCR?

Responses the Reviewer 1 Comment #10: Thank you for these critical comments. As reviewer’s comments, we revised the “PCR reaction” part [L108-110].

Reviewer 2 Report

Development of CAPS markers for evaluation of genetic diversity and population structure in the germplasm of button mushroom (Agaricus bisporus):  The experimental work appears to have been carried out well. However, a few points deserve attention for further publication. I suggest that it is accepted for publication after the following revisions:

- ABSTRACT: What parameters were optimized in Development of CAPS markers for evaluation of genetic diversity and population structure in the germplasm of button mushroom (Agaricus bisporus)? Authors must include numbers with the results found.

- CAPS markers = these markers should be better explored in the manuscript. How does it work? What is the mechanism of action?

- 4.1. Polymorphism did not differ according to SNP mutation type: Research suggests this is due to the higher number of possibly damaging non-synonymous changes resulting from Tv mutations compared with Ts mutations. Thus, Tv changes have a greater physi- cochemical impact on amino acid sequences and are not favored during natural selection: The impact on amino acid sequences and are not favored during natural selection: This information needs to be better explained in the manuscript.

- 4.2. Diversity of developed SNP markers: SSR and SNP markers exhibit different polymorphism traits, such as the number of  repeats in the sequence, mutations of a single nucleotide, and genome-wide mutations, and the ability to obtain different types of information in the same study may facilitate the  combined use of the two marker types. How do polymorphism traits work, what are these mechanisms?

 - 4.3. Population structure analysis using SNP markers: AMOVA of a model-based population revealed vari- 267 ation between groups and accessions by collection area (13% and 86%, respectively) and 268 genotype (15% and 85%, respectively). How can this difference be explained?

 Author Response

From

Ick-Hyun Jo

Department of Herbal Crop Research, National Institute of Horticultural and Herbal Science (NIHHS), Rural Development Administration (RDA), Eumseong 27709, Republic of Korea

[Tel.: +82-43-871-5613]

[Fax: +82-43-871-5639]

E-mail: intron@korea.kr

Jong-Wook Chung

Department of Industrial Plant Science and Technology, Chungbuk National University Cheongju

28644, Republic of Korea

[Tel.: +82-43-261-2524]

[Fax: +82-43-271-0413]

Email address: jwchung73@chungbuk.ac.kr

To

The Reviewer

Journal of Fungi

Dear Sir,

Sub: Regarding submission of the revised manuscript with id- jof-1200916-reg.

Herewith I am submitting our revised manuscript entitled of

Development of CAPS markers for evaluation of genetic diversity and population structure in the germplasm of button mushroom (Agaricus bisporus) manuscript id- jof-1200916 towards the publication in your prestigious journal- Journal of Fungi. As per your direction and reviewer recommendation, I assure you that all the corrections have been carried out according to reviewer’s comments and suggestions. I am looking forward to your favorable reply with optimism.

With kind regards,

Sincerely

Ick-Hyun Jo & Jong-Wook Chung

 Reviewer Comment and response

 Reviewer 2 Comment #1: ABSTRACT: What parameters were optimized in Development of CAPS markers for evaluation of genetic diversity and population structure in the germplasm of button mushroom (Agaricus bisporus)? Authors must include numbers with the results found.

Responses the Reviewer 2 Comment #1: Thank you so much for your valuable suggestion. We developed new CAPS markers for the analysis of the genetic diversity and population structure of Agaricus bisporus. In addition, in order to increase the efficiency of the analysis, the minimum marker combination was selected through the accumulation curve. The optimized parameter mentioned by the reviewer is considered to be the GD and MAF values used to select the minimum marker combination. Therefore, the optimized parameter of the GD and MAF values used to select the minimum marker combination was additionally written and reflected in the abstract. We have made additional modifications and highlighted with green color in the revised text [L32-34].

Reviewer 2 Comment #2: CAPS markers = these markers should be better explored in the manuscript. How does it work? What is the mechanism of action?

Responses the Reviewer 2 Comment #2: Thank you for these critical comments. We agree with your opinion.

In order to understand the purpose of this study and clarify the logic, an additional explanation of the mechanism of CAPS markers will be needed. Therefore, we reviewed the mechanism of CAPS markers and related research in the introduction session [L79-85].

 Reviewer 2 Comment #3: (4.1.) Polymorphism did not differ according to SNP mutation type: Research suggests this is due to the higher number of possibly damaging non-synonymous changes resulting from Tv mutations compared with Ts mutations. Thus, Tv changes have a greater physicochemical impact on amino acid sequences and are not favored during natural selection: The impact on amino acid sequences and are not favored during natural selection: This information needs to be better explained in the manuscript.

Responses the Reviewer 2 Comment #3: In response to your comments, additional explanations have been written in the discussion section. SNP mutations were described in more detail by referring to previous studies that Ts mutations, which do not affect amino acid translation, occur more frequently than Tv mutations [L250-254].

 Reviewer 2 Comment #4: (4.2.) Diversity of developed SNP markers: SSR and SNP markers exhibit different polymorphism traits, such as the number of repeats in the sequence, mutations of a single nucleotide, and genome-wide mutations, and the ability to obtain different types of information in the same study may facilitate the combined use of the two marker types. How do polymorphism traits work, what are these mechanisms?

Responses the Reviewer 2 Comment #4: SSR and SNP are mutations that generally occur and are found extensively within the genome, but have different characteristics. The differences in the polymorphism traits and mechanisms identified by SSR and SNP were additionally described, and the revised sentence was reflected in the discussion section (4.2) [L268-273].

 Reviewer 2 Comment #5: (4.3.) Population structure analysis using SNP markers: AMOVA of a model-based population revealed variation between groups and accessions by collection area (13% and 86%, respectively) and genotype (15% and 85%, respectively). How can this difference be explained?

Responses the Reviewer 2 Comment #5: As a result of AMOVA analysis, there was no significant difference between groups and accessions. We speculate that this phenomenon is the result of most commercially cultivated button mushrooms derived from similar strains. This is described in discussion section (4.3) L299-301. However, as it was explained that there was a difference in context, this part was corrected with a slight difference.
